# Performance Evaluation of the STANDARD i-Q COVID-19 Ag Test with Nasal and Oral Swab Specimens from Symptomatic Patients

**DOI:** 10.3390/diagnostics14020231

**Published:** 2024-01-22

**Authors:** Jong Do Seo, Hee-Won Moon, Eunju Shin, Ji Young Kim, Sang-Gyu Choi, Ju Ae Lee, Jeong Hwa Choi, Yeo-Min Yun

**Affiliations:** 1Department of Laboratory Medicine, Konkuk University Medical Center, Seoul 05030, Republic of Korea; 20210128@kuh.ac.kr (J.D.S.); 20210451@kuh.ac.kr (E.S.); 20050002@kuh.ac.kr (J.Y.K.); choice1182@kuh.ac.kr (S.-G.C.); ymyun@kuh.ac.kr (Y.-M.Y.); 2Department of Laboratory Medicine, Konkuk University School of Medicine, Seoul 05030, Republic of Korea; 3Department of Infection Control, Konkuk University Medical Center, Seoul 05030, Republic of Korea; 20060026@kuh.ac.kr (J.A.L.); jflower@kuh.ac.kr (J.H.C.)

**Keywords:** COVID-19, SARS-CoV-2 rapid antigen test, point-of-care testing, nasal swab, oral swab, performance evaluation

## Abstract

We evaluated the diagnostic performance of the STANDARD i-Q COVID-19 Ag Test, which was developed to detect viral antigens, using nasal and oral swabs. Sixty positive and 100 negative samples were analyzed. We determined the distribution of the Ct values according to the day of sample collection after symptom onset, the diagnostic performance of the total samples and subgroups separated by Ct value or time of sample collection, and the Ct value at which maximal accuracy was expected. No differences were observed in Ct values, except for the samples obtained on the day of symptom onset. The diagnostic sensitivity and specificity of the oral swabs were 75.0 and 100.0%, respectively, whereas those of the nasal swabs were 85.0 and 98.0%, respectively. The sensitivity was higher in samples with a high viral load collected earlier than those collected later, although the difference was not significant. False-negative results were confirmed in all samples with a Ct value ≥ 30.0. These results indicate that tests using oral and nasal swabs are helpful for diagnosing acute symptomatic cases with suspected high viral loads. Our tests exhibited relatively low sensitivity but high specificity rates, indicating the need to assess negative antigen test results.

## 1. Introduction

Since the first case report in Wuhan, China, in 2019, coronavirus disease 2019 (COVID-19), caused by the severe acute respiratory syndrome coronavirus-2 (SARS-CoV-2) infection, has rapidly spread to become a global pandemic, with 765 million reported cases worldwide as of April 2023 [1]. Molecular diagnostic tests that detect the viral genome in nasopharyngeal or oropharyngeal swab samples using a real-time reverse transcription polymerase chain reaction (rRT-PCR) are recognized as the gold standard for diagnosing SARS-CoV-2 infection [2,3,4]. However, molecular diagnostic tests have various limitations, such as long turnaround times (TATs), high costs, difficulties in the operation and maintenance of equipment, and the need for skilled personnel—issues that present a challenge for use in clinical labs [3,5,6,7,8].

Automated immunoassays [9,10,11,12,13] and rapid tests using immunochromatographic assays to detect viral antigens or antiviral antibodies have been developed [13,14,15] to overcome the limitations of molecular diagnostic tests. These tests have much shorter TATs and lower costs than other molecular diagnostic methods used for confirming diagnoses [7,8,13].

Antigen testing targeting the viral antigen itself rather than the reactive antibody is appropriate for diagnosing infections in environments with high antibody prevalence due to past infections or vaccinations [2,7,13,15]. It does not require any advanced professional equipment and is suitable for use in point-of-care testing (POCT); therefore, it can be used as an alternative to overcome the drawbacks of molecular confirmation tests [7,16,17]. 

Specific analytical performance standards must be met for the developed test to be used clinically. According to the World Health Organization (WHO) guidelines, to be approved, the rapid antigen tests should have sensitivity rates > 80% and specificity rates > 97% compared to nucleic acid amplification tests in suspected cases [17]. The Ministry of Food and Drug Safety of the Republic of Korea (KMFDS) requires the rapid antigen testing to have a sensitivity rate > 80% with a 95% confidence interval (CI) lower limit > 70% and a specificity rate > 95% with a 95% CI lower limit > 90% [18]. 

To date, various SARS-CoV-2 rapid antigen tests have been developed, approved, and used for diagnosis. These tests commonly use swab samples collected from the respiratory tract, such as the nasopharynx or oropharynx, and their performance is verified using standard samples [17]. However, discomfort during nasopharyngeal or oropharyngeal swab sample collection makes it difficult to obtain adequate samples. Therefore, the possibility of affecting the diagnostic performance cannot be ruled out. 

Antigen tests are known to play a limited role in the screening of asymptomatic patients because of their relatively low sensitivity and specificity rates [19]; thus, they are not recommended for these populations. This study evaluated the diagnostic performance of a rapid antigen test using oral and nasal cavity swab samples that are more representative of actual clinical settings than the approved nasopharyngeal or oropharyngeal swab samples [18] obtained under strictly controlled environments in patients with suspected SARS-CoV-2 infection.

## 2. Materials and Methods

### 2.1. Samples

The study enrolled patients with suspected symptoms when undergoing infection screening in the Konkuk University Medical Center triage room for SARS-CoV-2 [20] from May 2022 to September 2022. Participants were recruited after providing informed consent; oral and nasal swab samples were obtained for the study, in addition to nasopharyngeal swabs for routine molecular diagnostic tests. All procedures followed the principles of the Declaration of Helsinki after obtaining approval from the Institutional Review Board (IRB No. KUMC 2022-03-043).

As stated in the WHO guidelines, a clear deification about the study design is required because many factors, such as excluding asymptomatic patients from the study cohort or involving examiners who collect samples and perform tests, can affect the diagnostic performance [17]. The antigen test is recommended for symptomatic patients within 7 days after symptom onset (DASO) under specific conditions (high infection prevalence rate, delayed molecular confirmation test, and available effective prophylaxis) [8]. Adult patients (1) over the age of 19 who had (2) symptoms of suspected infection (such as fever, sore throat, shortness of breath, and pneumonia) and (3) within 7 days before the visit were included in the study after providing informed consent. (1) Patients who were younger than 19 years old, (2) asymptomatic individuals undergoing screening tests, (3) those who exceeded seven DASO at the time of visit when the day of symptom onset was considered day 0, (4) those who took antiviral drugs before the visit, and (5) those who did not submit written informed consent were excluded.

Following the manufacturer’s instructions, the nasal swab samples were collected by placing the swab into the nostrils (<1.5 cm) and rubbing the inside while rotating ten times on both sides; oral swab samples were collected by rubbing the tongue base and palate five times each. To ensure statistical significance and meet the diagnostic performance goal claimed by KMFDS, the sample size for this study was determined using the following formula:N = (Z_α/2_ + Z_β_)^2^ × P_1_(1 − P_1_)/(P_1_ − P_0_)^2^,
where P_1_ is the presumed performance (sensitivity or specificity) of the test; P_0_ is the confidence interval lower limit of the performance target; Z_α/2_ the z-statistic of the type I error at the significance level of α; and Z_β_ the z-statistic of the type II error with a statistical power of 1 − β. The minimum size of the positive study samples was 44.43, using 0.85 as P_1_ for sensitivity, 0.70 as P_0_, 1.96 as Z_α/2_ at a significance level of 0.05, and 0.84 as Z_β_ with a statistical power of 0.80. The negative samples were determined as 83.63, using 0.96 as P_1_ for specificity and 0.90 as P_0_.
N_pos_ = (1.96 + 0.84)^2^ × 0.85(1 − 0.85)/(0.85 − 0.70)^2^ = 44.43,
N_neg_ = (1.96 + 0.84)^2^ × 0.96(1 − 0.96)/(0.96 − 0.90)^2^ = 83.63.

Accordingly, we collected more than 60 positive samples containing at least a 10% low viral load and more than 100 negative samples. The samples from each group were analyzed in accordance with the order of samples taken. When 54 positive samples with C_t_ values smaller than 30 were collected, six positive samples with low viral loads with C_t_ values larger than 30 were available and 100 negative samples were completed to exclude selective bias.

### 2.2. Procedure

A molecular test was performed on a CFX 96 Real-Time Detection System (Bio-Rad, Hercules, CA, USA) using a Real-Q 2019-nCoV Kit (Biosewoom, Seoul, Republic of Korea). The cut-off C_t_ values of the *RdRp* and *E* genes were 38.0 for both. Cases with C_t_ values smaller than the cut-off for both genes were considered positive, whereas those with C_t_ values larger than the cut-off for both genes were considered negative. Cases with C_t_ values smaller than the cut-off for only one gene were considered indeterminate. 

The diagnostic performance of the rapid viral antigen test, the STANDARD i-Q COVID-19 Ag Test (SD Biosensor, Suwon, Republic of Korea), based on a rapid chromatographic immunoassay as the test principle, was evaluated in this study using oral and nasal swab samples. All tests were performed by following the manufacturer’s instructions using fresh samples on the collection day, and the results were interpreted and recorded by experts.

### 2.3. Statistical Analysis

The qualitative decision of the antigen test was compared with that of the molecular test, then the diagnostic performance parameters, sensitivity, specificity, positive predictive value (PPV), negative predictive value (NPV), and their 95% confidence intervals (CIs) were calculated for all samples, subgroups divided by C_t_ value, and subgroups divided by sample collection time. The cut-off C_t_ values for each gene in each sample type for which the maximal antigen test accuracy could be achieved were estimated using a receiver operating characteristic curve to determine the patient populations that could expect a high clinical value of antigen tests. Statistical analyses were performed using IBM SPSS Statistics for Windows (version 26.0; IBM Corp., Armonk, NY, USA) and MedCalc software (version 14.8.1; MedCalc Software, Ostend, Belgium).

## 3. Results

A total of 201 samples for the study were collected from May to September 2022. Two samples were excluded for exceeding the DASO criteria and other two were excluded with an ‘indeterminate’ decision; 197 samples were suitable for the analysis. There were 103 PCR-negative samples; eight of the remaining 94 PCR-positive samples had a low viral load (C_t_ > 30), while the other 86 samples showed C_t_ values smaller than 30. As mentioned above, a total of 160 samples consisting of 54 positive samples with C_t_ values lower than 30, six positive samples with low viral loads having C_t_ values larger than 30, and 100 negative samples were included in the final analysis, in order of sample collection.

No PCR-positive samples were obtained from patients who visited the hospital after four DASO. Among the positive samples, three were obtained from patients who visited on the day of symptom onset (DASO 0), and 32, 15, and 10 samples were obtained from patients with DASO 1, 2, and 3, respectively. The distribution of C_t_ values in 60 PCR-positive nasopharyngeal swab samples was analyzed on the day of sample collection. Significant differences were observed only between days 0 and 1 for the *RtRp* and *E* genes (Figure 1). The C_t_ values for both genes were significantly lower on day 1 than on day 0, suggesting that the viral load increased, whereas the change in C_t_ values from day 1 to day 3 was statistically insignificant, although they tended to increase, suggesting a decrease in the viral load.

When oral swabs were used as study samples for antigen testing, the sensitivity and specificity with 95% CI values were 75.0% (62.1–85.3%) and 100.0% (96.4–100.0%), respectively (Table 1). For the PPV and NPV, the values and their 95% CIs were 100.0% (92.1–100.0%) and 87.0% (79.4–92.5%), respectively, when the disease prevalence rate was assumed to be 37.5% (60/160), the positive rate of the study sample. The PPV and NPV were omitted from the table because of their dependence on ‘artificial’ prevalence in the present study design, which is not representative of the real disease prevalence rate.

For the nasal swab samples, the sensitivity, specificity, PPV, NPV, and their 95% CI values were 85.0% (73.4–92.9%), 98.0% (93.0–99.8%), 96.2% (87.0–99.5%), and 91.6% (84.6–96.1%), respectively.

When the diagnostic sensitivity and specificity of the antigen test in the subgroups were divided according to C_t_ value (≤20; 20 < ≤ 25; 25< ≤ 30; 30<), there was a tendency to show higher sensitivity in subgroups with lower C_t_ values, representing higher viral loads, although it was statistically insignificant, with the 95% CIs overlapping each other (Table 2). The diagnostic performances for the subgroups divided by the time of visit (DASO ≤ 1; 2–3) also exhibited no statistically significant difference (Table 3).

None of the PCR-positive patients with C_t_ values higher than a specific cut-off value were considered positive in the antigen test. These cut-off values were higher in the nasal swabs than in the oral swabs because one subject with a C_t_ value near but lower than 30 showed antigen-positive results in their nasal swabs and antigen-negative results in their oral swabs (Figure 2). In contrast, some participants showed negative antigen test results despite having a low C_t_ value in the PCR test, suggesting a sufficiently high viral load.

## 4. Discussion

Molecular tests are the most sensitive for detecting SARS-CoV-2 with a low viral load and can be used as confirmatory diagnostic tests; however, the shortcomings, such as the long TATs, high costs, and the necessity of expensive instruments and experienced examiners, limit their clinical use. Currently, various rapid SARS-CoV-2 viral antigen tests that can overcome the shortcomings of molecular testing have been developed. However, these rapid antigen tests are prone to false-positive or false-negative results because it is difficult to maintain the sensitivity and specificity of molecular tests [8,19]. Thus, approval authorities such as the WHO and KMFDS require antigen tests to meet the suggested sensitivity and specificity goals [17,18].

Although the tests launched in the market have passed the standard, the certified performance under strict controls may not be sufficient to reflect the performance in practical fields that need to consider the possibility of inappropriate sampling. Many factors, such as patient characteristics and the examiner’s proficiency, are known to affect the diagnostic performance. Therefore, we considered it meaningful to evaluate the performance under practical conditions using nasal or oral swab samples by ‘practitioner but not under a strictly controlled environment’, which represents the general condition of using rapid antigen tests.

In the present study, the diagnostic sensitivity with a 95% CI for rapid antigen tests using oral swab samples from symptomatic patients was 75.0% (62.1–85.3%), which did not meet the approval standard of the KMFDS (80.0%, 95% CI lower limit larger than 70.0%), although it showed 100% (96.4–100.0%) specificity, exceeding the standard (95.0%, 95% CI lower limit larger than 90.0%). When the sensitivity was evaluated in the subgroups divided by C_t_ value or DASO, the subgroups with C_t_ ≤ 25 met the goal; however, the C_t_ value could not be checked at the time during the antigen test. For the subgroups divided by DASO, it did not meet the standard in any subgroup, including DASO 1, which had a relatively higher viral load.

In contrast, the results using nasal swab samples showed 85.0% (73.4–92.9%) sensitivity and 98.0% (93.0–99.8%) specificity, meeting the KMFDS standard [18]. The performance in our study was in agreement with the results of a meta-analysis that included studies using the same test kits [21]. However, the influences of the C_t_ value and DASO between adjacent subgroups were insignificant in our study, contrary to previous studies that revealed a correlation between the C_t_ value, DASO, and clinical performance [21,22,23], possibly due to the differences in the study sample size and the criteria for dividing the subgroups for the analysis. In fact, another study with a design similar to ours showed that the effects of various factors on antigen test sensitivity were not statistically significant, as in our study [24]. The low sensitivity compared to other studies for low viral load samples could be considered to originate from the differences in the study samples and the selection of molecular tests.

As a result, unlike oral swabs, nasal swabs can be considered an alternative to nasopharyngeal swabs for infection screening. In addition, only some of the recruited samples were selected for the final analysis in this study to include low viral load (C_t_ > 30) samples over 10% of the total positive samples; thus, the ratio of low-load samples was adjusted to be higher than that recruited. Therefore, improved sensitivity can be expected in the clinical environment, given that the practical proportion of low viral load-positive samples in patients with DASO 7 is smaller than that in the present study. This result is consistent with another study that evaluated the utility of oral and nasal swabs as PCR test samples, in which oral swabs showed significant differences in C_t_ values compared with nasopharyngeal swabs, whereas nasal swabs did not [25].

When the distribution of the C_t_ values in the antigen test-positive and antigen test-negative groups was assessed, antigen tests were not expected to be useful in samples with low viral loads. The cut-off C_t_ values distinguishing the antigen-positive and antigen-negative groups were slightly higher in the nasal swabs than in the oral swabs for both genes; therefore, the sensitivity of the nasal swabs appeared to be slightly higher than that of the oral swabs. Some samples showed false-negative results despite a high viral load (low C_t_ value); these samples were possibly higher in the oral swabs than in the nasal swabs (eight samples vs. three samples), possibly due to the heterogeneous viral distribution depending on the sample collection site (nasopharyngeal swab vs. nasal or oral swab); the nasal cavity seemed to better reflect the distribution of viruses in the respiratory tract than in the oral cavity. Therefore, attention should be paid to the interpretation of antigen test results using nasal and oral swabs; it should be kept in mind that negative results in rapid antigen screening tests are not sufficient evidence to rule out infection, particularly when the test has been performed too early or late after symptom onset or when oral swabs are used. Even if the antigen screening test results are negative, it is necessary to perform confirmatory molecular tests and provide empirical treatment for infections when the disease is clinically suspected.

Considering the C_t_ value distribution according to DASO, antigen tests performed at an early infection stage with a high viral load are expected to be helpful; however, interpretations of the samples obtained on the day of symptom onset (DASO 0) should be cautious because of the high C_t_ values.

However, these findings were demonstrated in a single assay, the STANDARD i-Q COVID-19 Ag Test, which uses a lateral flow immunoassay as the principle, and the samples were collected by imitating the technique commonly used by the general Korean population who were not familiar with self-testing for infectious diseases and sample collection until the COVID-19 outbreak. Therefore, it is necessary to remember that our findings are not representative of all SARS-CoV-2 antigen tests that use other principles or perform specific preparations for oral samples.

This study had some intrinsic design limitations. First, the total number of study subjects, determined following the diagnostic performance evaluation guidelines suggested by the KMFDS, was not sufficient to secure statistical power for subgroups classified based on DASO and C_t_ values because the sample size was set for the entire group. Therefore, there may be differences in the influence of each factor or statistical significance as the sample size changes. Second, this study was conducted by skilled examiners who perform diagnostic procedures in hospital triage rooms during sample collection and result reading; therefore, the study design may not sufficiently reflect the practical tests conducted by less-skilled persons, such as the patients themselves, their family members, their friends, and primary care staff. Third, the possibility of lower sensitivity in asymptomatic cases than in patients with suspected symptoms should be considered, since the results of the present study were determined from patients with suspected infection symptoms within 7 days before the hospital visit, according to the guidelines. Lastly, because this study was designed to evaluate the performance of antigen tests in practical clinical settings such as a ‘semi-quantitative’ reporting format of a PCR test, the variables such as the viral genotype, ‘quantitative’ C_t_ value in the molecular test, and performance comparison with other rapid antigen tests, which are not used routinely in the clinical field but are expected to provide additional information for research, were not evaluated.

Despite some limitations, this study simulates a “less strictly controlled” point-of-care test environment in which rapid viral antigen tests are commonly performed and reveals the diagnostic performance using nasal and oral swab samples. The conditions recommended for screening—clinical symptoms within 7 days before the visit—should be understood for the practical clinical use of antigen tests. The rapid antigen test using nasal swabs revealed in this study is expected to help understand the clinical usefulness of POCT and the interpretation of the results. Improvements in the study design, such as varying the test conditions from sample collection to preparation, can be considered in future studies.

## Figures and Tables

**Figure 1 diagnostics-14-00231-f001:**
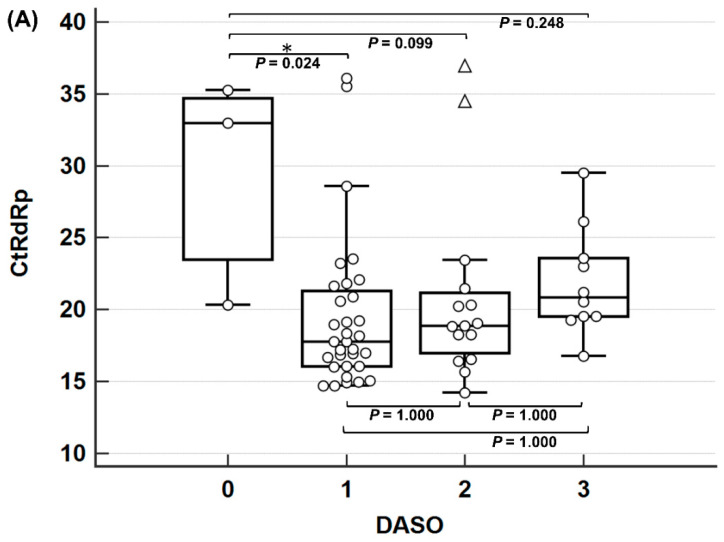
Distribution of C_t_ values for (**A**) *RtRp* and (**B**) *E* genes in the polymerase chain reaction (PCR)-positive samples compared using sample collection days after symptom onset and using an analysis of variance (ANOVA) with a Bonferroni post hoc test. The *p*-values from the Bonferroni test are presented. The circles represent the C_t_ values of each subject, triangle represent extreme outliers confirmed by Tukey’s methods, and asterisk represent the subgroups that showed significant difference with *p*-value < 0.05. Abbreviation: DASO, days after symptom onset.

**Figure 2 diagnostics-14-00231-f002:**
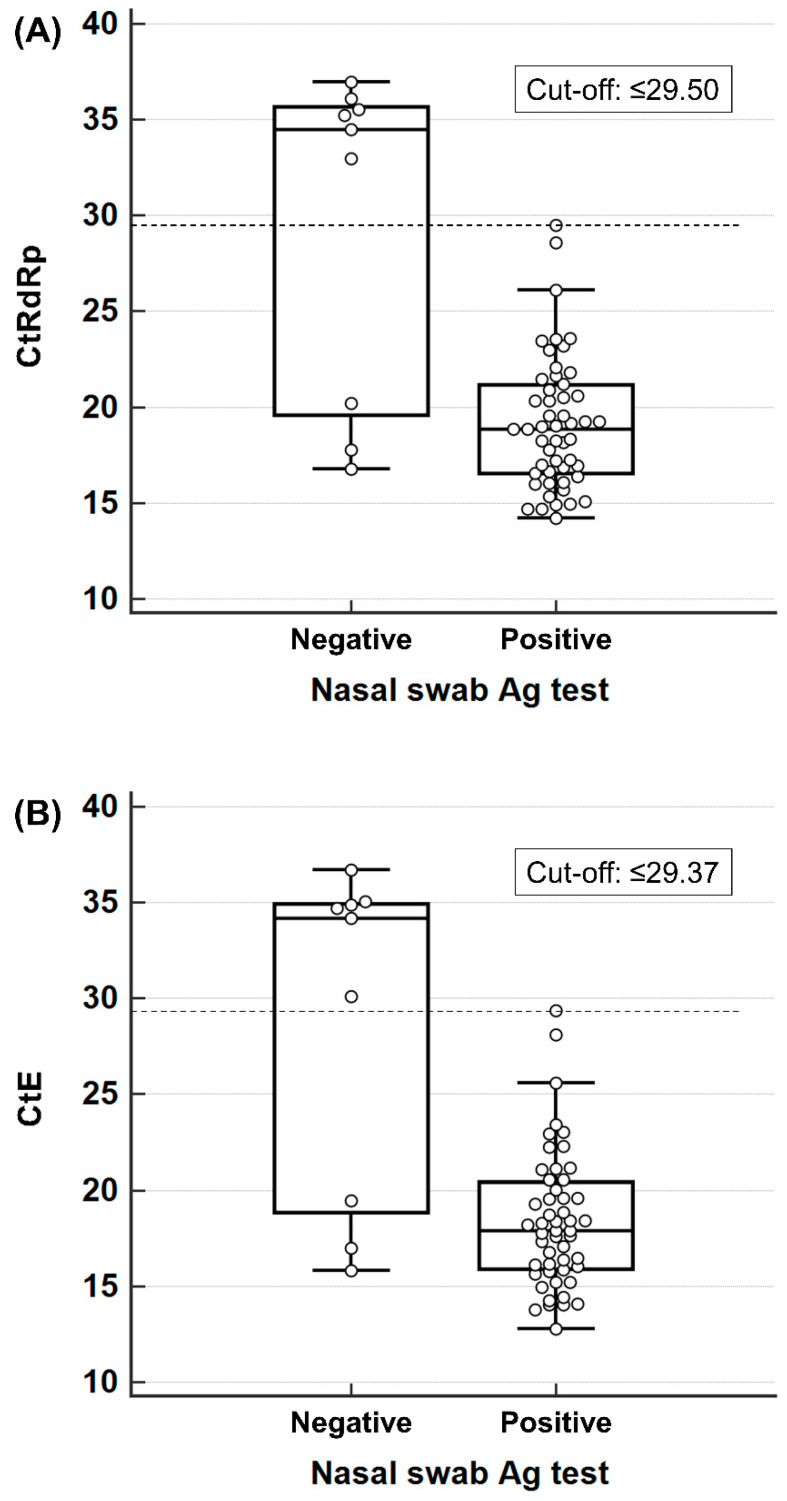
Distribution of C_t_ values and maximal antigen test sensitivity limits for (**A**) *RtRp*, (**B**) *E* in the nasal swab samples, (**C**) *RtRp*, and (**D**) *E* in the oral swab samples. The circles represent the C_t_ values of each subject.

**Table 1 diagnostics-14-00231-t001:** Diagnostic sensitivity and specificity of the antigen test using oral and nasal swab samples, calculated using a nasopharyngeal swab sample polymerase chain reaction (PCR) test as the gold standard.

	PCR+ (*n* = 60)		PCR− (*n* = 100)		Sensitivity (95% CI)	Specificity (95% CI)
	Ag+	Ag−	Ag+	Ag−		
Oral swab	45	15	0	100	75.0 (62.1–85.3)	100.0 (96.4–100.0)
Nasal swab	51	9	2	98	85.0 (73.4–92.9)	98.0 (93.0–99.8)

Abbreviation: CI, confidence interval.

**Table 2 diagnostics-14-00231-t002:** Diagnostic sensitivity and specificity of the antigen test based on the C_t_ value.

	PCR+		PCR−		Sensitivity (95% CI)	Specificity (95% CI)
	Ag+	Ag−	Ag+	Ag−		
Oral swab						
Ct*_RdRp_*						
≤20	30	5			85.7 (69.7–95.2)	
20 < ≤ 25	13	3			81.3 (54.4–96.0)	
25 < ≤ 30	2	1			66.7 (9.4–99.2)	
30<	0	6	0	100	0.0 (0.0–45.9)	100.0 (96.4–100.0)
Ct*_E_*						
≤20	34	6			85.0 (70.2–94.3)	
20 < ≤ 25	9	2			81.8 (48.2–97.7)	
25 < ≤ 30	2	1			66.7 (9.4–99.2)	
30<	0	6	0	100	0.00 (0.0–45.9)	100.0 (96.4–100.0)
Nasal swab						
Ct*_RdRp_*						
≤20	33	2			94.3 (80.8–99.3)	
20 < ≤ 25	15	1			93.8 (69.8–99.8)	
25 < ≤ 30	3	0			100.0 (29.2–100.0)	
30<	0	6	2	98	0.0 (0.0–84.2)	94.2 (87.9–97.9)
Ct*_E_*						
≤20	37	3			92.55 (79.6–98.4)	
20 < ≤ 25	11	0			100.0 (71.5–100.0)	
25 < ≤ 30	3	0			100.0 (29.2–100.0)	
30<	0	6	2	98	0.0 (0.0–45.9)	98.0 (93.0–99.8)

Abbreviation: CI, confidence interval.

**Table 3 diagnostics-14-00231-t003:** Diagnostic sensitivity and specificity of the antigen test based on the sample collection time.

	PCR+		PCR−		Sensitivity (95% CI)	Specificity (95% CI)
	Ag+	Ag−	Ag+	Ag−		
Oral swab						
DASO						
≤1	27	8	0	50	77.1 (59.9–89.6)	100.0 (92.9–100.0)
2–3	18	7	0	41	72.0 (50.6–87.9)	97.6 (87.4–99.9)
4–6			0	9		100.0 (66.4–100.0)
Nasal swab						
DASO						
≤1	30	5	1	49	85.7 (69.7–95.2)	98.0 (89.4–99.9)
2–3	21	4	0	41	84.0 (63.9–95.5)	100.0 (91.4–100.0)
4–6			1	8		88.9 (51.8–99.7)

Abbreviations: DASO, days after symptom onset; CI, confidence interval.

## Data Availability

The dataset presented in this article is available on request from the authors, although some part of information may not be available due to privacy of subjects and commercial restriction.

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
