# Peer review of "Performance Evaluation of the STANDARD i-Q COVID-19 Ag Test with Nasal and Oral Swab Specimens from Symptomatic Patients"

_diagnostics, 2024, doi:10.3390/diagnostics14020231_

Round 1

Reviewer 1 Report

Comments and Suggestions for Authors

In order to estimate the precision of the estimates and the power of the study, it is necessary to calculate a sample size.

Author Response

The number of study sample was determined in accordance with the guideline of Ministry of Food and Drug Safety of the Republic of Korea (KMFDS) for COVID-19 IVD approval to secure the statistical power of the evaluation. The formula and calculation process were added to the 2.1 sample section of the revised manuscript, from line 96 to 108. The calculation processes are as follow.

 Npos = (1.96+0.84)2×0.85(1-0.85)/(0.85-0.7)2 = 44.43

 Nneg = (1.96+0.84)2×0.96(1-0.96)/(0.96-0.9)2 = 83.63

Reviewer 2 Report

Comments and Suggestions for Authors

The manufacturer of the Test in question are a highly respected developer of rapid tests, so my assumption is the performance of the Test using oral swabs may be approved by some suggestions for any future work.

1) There are quite a number of studies on antigen tests that reflect equivalent or better performance than nasal swabs and this may be a consequence of the chemistry of the lateral flow test used, so maybe the authors can highlight this point.

2) Method of collection, particularly for oral samples is critical.  OraSure's OraQuick HIV Test is approved for OTC use in the US using a swab that collects around the gum line and other tests use a similar principle.  Experimentation with how the actual sample is collected could result in improved results for oral swab testing.

3)  Oral samples contain various interfering factors including mucins that can confound tests.  Saliva [purified] is another option to evaluate.

Comments on the Quality of English Language

Only small grammatical corrections needed [minor]

Author Response

- Our study had designed to simulate clinical setting in which rapid antigen test are commonly used, and it is not representative to ‘optimal’ condition. The description about this issue was added to revised manuscript, in discussion section from line 267 to 263, and clarified that the findings of present study are to evaluate the performance of a single antigen test, and are not representative of all kinds of POCTs.

 “However, these findings were demonstrated in a single assay, the STANDARD i-Q COVID-19 Ag Test, which uses lateral flow immunoassay as a principle, and the samples were collected by imitating the technique commonly used by the general Korean population not familiar with self-testing for infectious diseases and sample collection until the COVID-19 outbreak. Therefore, it is necessary to remember that our findings are not representative of all SARS-CoV-2 antigen tests that use other principles or perform specific preparations for oral samples.”,

And the description about the improvements in research design that are expected for future studies, are added in line 298-300,

“Improvements in the study design, such as varying test conditions from sample collection to preparation, can be considered in future studies.”

- Grammatical corrections was performed by English editing services.

Reviewer 3 Report

Comments and Suggestions for Authors

Seo et al. have assessed the diagnostic accuracy of a COVID-19-antigen test with a limited number of nasal and oral swabs compared to real-time PCR with semi-quantification based on cycle-threshold values. The test performance of the assay is comparable to similar assays published in dozens of publications before. Altogether, the scientific value of the work is low, as the authors abstained from performing any additional assessments that might have made the work scientifically interesting. In particular, the authors didn’t provide genomic variant information on the PCR-positive sample, they also didn’t provide absolute quantification of real-time PCR results but just Ct-value based semi-quantification and further, epidemiological information on the assessed patients and the disease stage (e.g., presumed time since the likely infection incident, etc.) is not shown. Not even head-to-head comparisons with other similar antigen-assays were included. At the beginning of the pandemic, such simple research work might have been justified to guide the pandemic management. Nowadays, however, after WHO has ceased the emergency status of the COVID-19 pandemic, I cannot recommend for publishing such a very simple study in a well-recognized, peer-reviewed international journal like MDPI diagnostics.

Author Response

- The authors agree with the reviewer's opinion that it is difficult to have a large clinical value at present, unlike the early stage of pandemic. This study was designed in early 2022, when the pandemic was at peak in Korea, and was performed to utilize the result as information for clinical use of rapid antigen tests. Even if it does not have great academic significance at present, we thought that the influence of sample types can be used as information for evaluating the performance of other viral antigen tests in the future. Since the performance of actual clinical use was evaluated, information not used in clinical setting was not included for the evaluation, and the description about this issue was added in the revised manuscript as limitation, in line 286-291.

“Lastly, because this study has been designed to evaluate the performance of antigen tests in practical clinical settings such as a ‘semi-quantitative’ reporting format of PCR test, the variables like viral genotype, ‘quantitative’ Ct value in molecular test, and performance comparison with other rapid antigen tests, which are not used routinely in the clinical field though expected to provide additional information for research, were not evaluated.”

Round 2

Reviewer 3 Report

Comments and Suggestions for Authors

No additional comments.